# Effect of Oleoylethanolamide-Based Dietary Supplement on Systemic Inflammation in the Development of Alimentary-Induced Obesity in Mice

**DOI:** 10.3390/nu15204345

**Published:** 2023-10-12

**Authors:** Darya Ivashkevich, Arina Ponomarenko, Igor Manzhulo, Ruslan Sultanov, Inessa Dyuizen

**Affiliations:** A.V. Zhirmunsky National Scientific Center of Marine Biology, Far Eastern Branch, Russian Academy of Sciences, 690041 Vladivostok, Russia; owncean@yandex.ru (D.I.); i-manzhulo@bk.ru (I.M.); sultanovruslan90@yandex.ru (R.S.); duval@mail.ru (I.D.)

**Keywords:** OEA, oleoylethanolamide, obesity, inflammation, spleen, cytokines

## Abstract

The complex effect of oleoylethanolamide-based dietary supplement (OEA-DS) was studied in a model of diet-induced obesity in mice. Physiological, biochemical, and immunohistochemical methods were used to reveal differences in the changes in the weight of experimental animals, morphological changes in the spleen tissues, and changes in the cytokine expression profile in the spleen, blood plasma, and macrophage cell culture. First, it is shown that a hypercaloric diet high in carbohydrates and cholesterol led to the development of systemic inflammation, accompanied by organ morphological changes and increased production of proinflammatory cytokines. In parallel, the use of OEA-DS reduced the intensity of cellular inflammatory reactions, accompanied by a decrease in markers of cellular inflammation and proliferation, such as CD68, Iba-1, and Ki67 in the spleen tissue, and stabilized the level of proinflammatory cytokines (IL-1β, IL-6, TNFα) both in animals and in cell culture. In addition, in the macrophage cell culture (RAW264.7), it was shown that OEA-DS also suppressed the production of reactive oxygen species and nitrites in LPS-induced inflammation. The results of this study indicate the complex action of OEA-DS in obesity, which includes a reduction of systemic inflammation.

## 1. Introduction

The high incidence of obesity and its associated metabolic and somatic disorders (diabetes mellitus, atherosclerosis, cardiovascular pathology) dictate the need to identify new molecular mechanisms involved in the complex and systemic regulation of these pathophysiological processes [1]. Despite the high prevalence of obesity, methods of dietary and pharmacological therapy are presently limited [2,3,4,5]. Among dietary methods of correction, in addition to classical caloric restriction, time-restricted eating (TRE) protocols are also used, which are periods of eating separated by periods of fasting (from 12 h to 48 h or more). The results of clinical trials in most cases confirm the effectiveness of TRE for weight loss [6,7,8]; however, this method also has its limitations [9,10]. Prescription drugs studied in large, randomized, placebo-controlled clinical trials and approved for the long-term treatment of obesity include phentermine, an appetite suppressant that increases catecholamine levels and induces a feeling of satiety; orlistat, a gastrointestinal lipase inhibitor that reduces fat absorption; and sibutramine, a noradrenaline–serotonin–dopamine reuptake inhibitor that suppresses appetite and may also increase energy demand [2]. The usage of these drugs is complicated by serious gastrointestinal (orlistat) or cardiovascular (sibutramine) side effects [3], which often ultimately lead to polypragmasy. Meanwhile, glucagon-like peptide-1 receptor agonists (liraglutide, semaglutide) hold great promise for treatment [11]. However, because of their rapid degradation in the stomach, they are usually administered subcutaneously [12]. Although GLP-1 receptor agonists have some side effects, they are considered less significant than other anti-obesity drugs [13,14,15]. Moreover, the pharmacotherapy results exceed the effects achieved by simple dietary restrictions and physical activity by only a small percentage.

In this regard, the development of fundamentally new treatments for obesity is a priority for both patients and the health care system. The most promising pharmacological targets for the development of drugs with metabolic and anorexigenic effects can be elements of human and animal bodies’ own endocannabinoid systems, which are represented by a group of lipid nature compounds—N-acylethanolamides (NAEs) of saturated and polyunsaturated fatty acids [16]. In the present work, the essential fatty acids of olive oil were used as a source for OAE synthesis. In general, extra virgin olive oil (EVOO) consists predominantly of monounsaturated fatty acids (MUFAs), with oleic acid (C18:1) of 55% to 83%, followed by polyunsaturated fatty acids (PUFAs) of 4% to 20%, mainly linoleic acid (C18:2), and saturated fatty acids (SFAs) of 8% to 14%, such as palmitic acid (C16:0) [17,18,19]. The accumulated data suggest that OEA may be involved in various pathophysiological aspects of appetite regulation and lipid and carbohydrate metabolism. This endogenous metabolite is synthesized in the upper small intestine during lipid absorption, and it modulates the expression of genes involved in fat absorption and fatty acid metabolism [20]. Due to its wide range of biological effects, OEA has great potential in the pharmacotherapy of obesity and associated lipid disorders [21,22]. OEA administration appears to reduce appetite and inhibit weight gain in both lean and obese rodents [23]. These effects of OEA are mediated primarily by the activation of the peroxisome proliferator-activated receptor-α (PPAR-α) [24]. Other close structural analogs of OEA, such as linoleylethanolamide (LEA) and palmitoylethanolamide (PEA), also have a high affinity for PPAR-a and show similar anti-inflammatory properties [25]. However, their anorexic effects are very inferior to those of OEA. In terms of reducing food intake, PEA is significantly less effective, LEA is similar in potency to OEA, and oleic acid itself has no effect [26]. Nevertheless, endogenous levels of OEA, PEA, and LEA are regulated by the nutritional status of the animals. Fasting generally reduces NAE levels in the initial small intestine, while food intake stimulates duodenal and jejunal mucosal cells to produce endogenous OEA [27], PEA, and LEA [26,28]. Dietary fat content can also influence the endogenous intestine levels of OEA, LEA, and PEA in a time- and dose-dependent manner, which suggests their potential role as so-called fat sensors in the intestine [26]. Moreover, a diet high in sucrose and low in fat may reduce the mobilization of OEA and LEA from the small intestine during intraduodenal lipid administration [28]. These observations suggest that changes in the NAE pathways also play a role in lipid metabolism. It is now known that obesity is accompanied by a state of chronic systemic inflammation associated with elevated levels of circulating cytokines [29]. Serum IL-6 is increased in diabetics and non-diabetic obese people compared to non-obese controls [30,31]. The levels of other inflammatory cytokines, such as TNFα, MCP-1, IL-1β, and IL-8, also tend to increase in obesity [32]. With the development of systemic inflammation, an immune response usually occurs, accompanied by corresponding molecular and cellular changes in the organs of the immune system. The thymus and spleen are the main reservoirs for T lymphocytes, which can regulate the innate immune response and provide protection against pathogens and tissue damage. Oxidative stress and inflammation that develop in obesity, among others, can lead to the dysfunction of these organs [33]. The spleen size has been shown to increase in obese individuals [34]. Recently, an increase in spleen volume, a marker of chronic inflammation and immune system activation, has been shown to correlate with the progression of fatty liver dystrophy [35,36]. Finally, elevated levels of proinflammatory cytokines detected in the spleen, particularly IL-6, were found to be directly proportional to BMI and waist circumference [37]. Given the available data, we can assume that endogenous NEAs, including OEAs that interact with the PPAR-α receptor, help maintain homeostasis by preventing the triggering of systemic inflammation. It can be supposed that OEA, having a multifactorial complex effect on both the metabolic and immune processes, may be a promising candidate for the pharmacological adjustment of alimentary disorders and concomitant pathologies of carbohydrate and lipid metabolism.

In this study, an OEA-based dietary supplement (OEA-DS) was shown to have anorexic effects in a model of alimentary-induced obesity in mice, and its effects were accompanied by a decrease in systemic inflammation.

## 2. Materials and Methods

### 2.1. Production of a Dietary Supplement

A dietary supplement based on oleoylethanolamide was prepared from olive oil containing 72.3% oleic acid (Figure 1a). Methyl esters were isolated from olive oil by methanolysis in the presence of a sodium methoxide as an alkaline catalyst.

The methyl esters of oleic acid were isolated by HPLC on a Shimadzu LC-8A chromatograph (Shimadzu, Kyoto, Japan) with UV/VIS SPD-20A (205 nm). Separation was carried out on a Supelco Discovery HS C-18 preparative reverse-phase column (Bellefonte, PA, USA), with a 10 μm particle size, and 250 mm × 50 mm i.d. Isocratic elution with the system’s ethanol/water (90:10, *v*/*v*) was used. The elution rate was 50 mL/min. Fractions containing oleic acid ME (OAME) were collected, evaporated under vacuum, and analyzed by GC.

Fatty acid esters were analyzed by gas chromatography on a Shimadzu GC-17A chromatograph (Kyoto, Japan) with a flame ionization detector and a 30 m × 0.25 mm i.d. capillary column, the 10 (Supelco, Bellefonte, PA, USA). The analysis was performed under the following conditions: the column temperature was 205 °C, the temperature of the injector and detector was 250 °C. Helium was used as a carrier gas. The peaks of the fatty acid methyl esters were determined from the retention times of the individual FA esters by comparing their equivalent carbon length numbers. 

From the isolated oleic acid methyl ester of 97% purity, ethanolamide was prepared according to the Farris method [38]. The esters and ethanolamine were taken at a ratio of 1:1.1 (mol/mol), sodium methoxide was used as a catalyst, and the reaction proceeded at 90 °C for 4 h under reduced pressure. OEA was separated from the unreacted methyl esters and fatty acids by twofold precipitation in ethyl acetate at 0 °C (Figure 1b).

To determine the composition of ethanolamides, conversion to trimethylsilyl derivatives (TMS-NAE) was used [39]. For this, 50 µL of N, O-bis (trimethylsilyl) trifluoroacetamide (BSTFA) was added to 1 mg of fatty acid ethanolamides and heated to 60 °C for 1 h under argon. Then, to quantify the composition of ethanolamides, 1 mL of hexane was added, and 1 µL of each silylated fraction was injected into the GC system. A Shimadzu GC-2010 plus chromatograph with a Supelco SLB™ with a 5 ms capillary column 30 m × 0.25 mm inner (Sigma-Aldrich, Bellefonte, PA, USA) was used as well as a flame ionization detector (Shimadzu, Kyoto, Japan). The following conditions were applied to separate the components of the mixture: an initial temperature of 180 °C; a heating rate from 2 °C/min to 260 °C; and the temperature was maintained for 35 min. The injector and detector temperatures were the same and amounted to 260 °C. The composition of the obtained fatty acid ethanolamides was completely consistent with the original FAME (Figure 1c). 

### 2.2. Experimental Model

The experiment was performed on three-month-old female C57BL/6 mice. The animals were divided into 4 groups, with 12 animals in each group: “CTL”, the group receiving a standard diet; “CTL + OEA-DS”, the group receiving a standard diet with OEA-DS added at a dosage of 200 mg/kg per day orally along with food; “DIO”, the group of animals with diet-induced obesity; and “DIO + OEA-DS”, the group of animals with diet-induced obesity receiving OEA-DS at a dosage of 200 mg/kg per day. The animals were divided into 5 individuals from 1 group per cage.

The “CTL” and “CTL + OEA-DS” groups received a ready-made feed containing grains, high-protein components (vegetable and animal proteins), vegetable oil, amino acids, organic acids, vitamin-mineral complex (Delta Feds—C-19, Novosibirsk, Russia). The animals of the groups with induced obesity received modified feed containing 60% standard feed + 20% sucrose + 20% sunflower oil + 10% cholesterol. The weight measurements were taken once a week. The daily caloric intake was assessed once every 4 days by weighing the remaining mass of feed with further recalculation to its caloric value (the caloric value of standard feed was 252 kcal/100 g, the modified feed was 410 kcal/100 g) (Table 1). The period of the experiment was 2 months. At the end of the experiment, parameters such as the total weight gain over the entire period and the average daily caloric intake per animal were evaluated.

The excision of material for subsequent immunohistochemical study was performed at the end of the experiment, i.e., after 2 months from the beginning of the diet. For this purpose, the mice were anesthetized (3% isoflurane in 100% oxygen solution) using a vaporizer for rodent anesthesia (VetFloTM, Kent Scientific Corporation, Torrington, CT, USA). Then, the chest cavity was opened, blood was taken from the left ventricle using a syringe, and the biomaterial was sampled. To collect the material for histological and immunohistochemical studies, some animals underwent transcardial perfusion with cold 0.1 M phosphate buffer (PBS), pH 7.2, and then with fixative solution (10% formalin in 0.1 M PBS, pH 7.2). The spleen was then immediately excised and placed into freshly buffered 10% formalin for 24 h at 4 °C. After washing 5–6 times with PBS, the biomaterial samples were dehydrated and transferred into paraffin according to the standard protocol. Blood plasma and spleen for Western blotting were extracted without preperfusion and immediately frozen in liquid nitrogen, with further storage at −80 °C.

### 2.3. Immunohistochemical Staining

Paraffin sections of the spleen (7 μm) were dewaxed, then incubated in 3% hydrogen peroxide for 15 min to block endogenous peroxidase. This was followed by 3 washes with 0.1 M phosphate buffer (pH 7.2), each for 10 min. The sections were subsequently incubated for 1 h in blocking buffer (PBS, 2% BSA (SC-2323, Santa Cruz Biotechnology, Santa Cruz, CA, USA), 0.1% Tween20, 0.25% Triton X-100 (Sigma, St. Louis, MO, USA). Next, without washing, primary mono- and polyclonal rabbit antibodies dissolved in blocking buffer were added to the next markers: Iba-1 (1:2000, ab108539, Abcam, Cambridge, MA, USA) CD68 (1:1000, ab125212, Cambridge, MA, USA), PPAR-α (1:1000, ab245119, Cambridge, MA, USA), CD163 (1:1000, ab182422, Cambridge, MA, USA), Ki67 (1:1000, PA5-19462, Thermo Fisher Scientific, Waltham, MA, USA). Sections with primary antibodies were incubated in a wet chamber in a refrigerator at 4 °C overnight. A negative control (without primary antibodies) was also incubated in parallel. The sections were subsequently washed with phosphate buffer (0.1 M, pH 7.2) 3 times for 10 min, after which the sections were incubated in a solution of secondary antibodies (Biotinylated Goat Anti-Rabbit IgG, ab178846Abcam, Cambridge, MA, USA) for 15 min. After 3 washes with buffer (0.1 M, pH 7.2), the sections were incubated for 10 min with streptavidin (ab64269, Abcam, Cambridge, MA, USA), and then washed again 3 times with phosphate buffer (0.1 M, pH 7.2). Then, the sections were processed with chromogen (Nova Red, Vector Laboratories, Burlingame, CA, USA) for 5 min. The sections were washed with distilled water, then dehydrated and mounted under glass slides using mounting medium. Immunohistochemical staining area was assessed using the ImageJ 1.41 software package (NIH, Bethesda, MD, USA).

Every fifth serial section was used for immunohistochemical staining. The area of the immunohistopositive zone was assessed using the ImageJ 1.41 software package (NIH, USA). In the case of the Iba-1, CD68, CD163, and PPAR-α markers, the ratio of the area of interest to the total area of immunopositive staining was assessed, which was expressed as a percentage. For the Ki67 marker, the number of immunopositive cells/mm^3^ was measured, which was calculated according to the formula:D = (10^9^ × n)/(S × 7), 
where D is the cell density; 10^9^ is the conversion factor of μm^3^ to mm^3^; n is the number of immunopositive cells; S is the area of the area under study (μm^2^); 7 is the slice thickness (μm). 

### 2.4. Histological Staining

For histological staining, the deparaffinized sections were placed for 5 min in Mayer’s hematoxylin solution (BioVitrum, Saint Petersburg, Russia), and then washed for 15 min under running water, followed by eosin staining (BioVitrum, Saint Petersburg, Russia). The sections were placed in a concentrated eosin solution for 5 s, then washed three times in 96% ethanol. After that, the sections were transferred into xylene for 10 min and then covered with slides using mounting medium.

### 2.5. Cell Culture

The mouse macrophage cell line RAW264.7 was used for in vitro studies. The cells were cultured in standard DMEM medium (Thermo Fisher Scientific, Waltham, MA, USA) containing 10% FBS (Thermo Fisher Scientific, Waltham, MA, USA), and 0.5% penicillin/streptomycin (Thermo Fisher Scientific, Waltham, MA, USA) at 37 °C in a humidified atmosphere with 5% CO_2_. Trypsin-EDTA (0.05%) (Thermo Fisher Scientific, Waltham, MA, USA) was applied for cell passaging. For the cytotoxicity studies, cells were seeded in 96-well microplates (1 × 10^3^ cells/well); for Western blotting, cells were placed in 6-well plates (1 × 10^5^ cells/well). Each experiment was performed independently three times.

### 2.6. MTS-Test

The cytotoxic activity of OEA-DS was assessed using the MTS test. Cells were plated in transparent 96-well plates (1 × 10^3^ cells/well) and incubated in standard medium for 1 h. After cell adhesion, the medium was replaced with OEA-DS solution at concentrations of 0.01, 0.1, 1, and 10 µg/mL and incubated overnight. Cells cultured in standard medium were used as a negative control. The next day, MTS reagent (ab197010, Abcam, Cambridge, MA, USA) was added to each well and incubated for 2 h at 37 °C. The optical density was then measured on a microplate reader (BioRad, Hercules, CA, USA) at 490 nm. The results are presented as percentages relative to the negative control.

### 2.7. Analysis of Reactive Oxygen Species (ROS) and Nitric Oxide (NO)

The antioxidant activity of OEA-DS was studied in a culture of macrophage cells activated by bacterial lipopolysaccharide (LPS). The experiments were carried out in 96-well plates, where cells were plated (1 × 10^3^ cells/well). At 1 h after cell adhesion, the culture medium was replaced with a medium supplemented by OEA-DS at concentrations of 0.001, 0.01, 0.1, 1, and 10 µg/mL. Then, LPS (1 mg/mL) was added to some wells to activate the cells. The cells activated only by LPS served as a positive control, and the cells in a standard culture medium served as a negative control. ROS activity was studied after 24 h of incubation by adding 10 µg/mL to 20 μL of 2,7-dichlorodihydrofluorescein diacetate solution (D399, Invitrogen, Carlsbad, CA, USA) according to the manufacturer’s recommendations. A 10 μm solution of DAF-FM diacetate (D23844, Thermo Fisher Scientific, Waltham, MA, USA) was used for NO level analysis according to the manufacturer’s recommendations. The fluorescence intensity was measured using a SPARK TECAN 10M tablet reader for ROS at λex/λem = 485/518 nm and for NO at λex/λem = 460/524 nm. The results obtained are presented as a percentage relative to the negative control.

### 2.8. Western Blotting

Macrophage culture was placed into 6-well plates at 1 × 10^9^ cells per well and divided into 4 groups: intact cells, cells treated with LPS at 1 μg/mL, and groups of cells treated with OEA-DS at concentrations of 1 and 10 μg/mL. After 24 h of incubation, the cells were sonicated homogenized in phosphate buffer (7.2 pH) with the addition of 150 mM serine protease inhibitor (PMSF, Helicon GC207002, Moscow, Russia). The spleen tissue was homogenized manually by pestle grinding. The plasma was not homogenized. The protein concentration in the solution was then measured, followed by equalization to a concentration of 2 mg/mL for the spleen tissue and cell culture and 1 mg/mL for the plasma. Then, the samples were diluted 1:1 with 1× sample buffer (BioRad, Hercules, CA, USA) containing 5% 2-mercapnoethanol (Sigma-Aldrich, M6250, St. Louis, MI, USA), followed by incubation for 5 min in a water bath at 94 °C. Electrophoresis was performed using ready-to-use gel cartridges (Any kDa Mini-PROTEAN gel, BioRad, Hercules, CA, USA) and a molecular ladder (Spectra Multicolor Broad Range Protein Ladder, Thermo Fisher Scientific, Waltham, MA, USA) in a BioRad chamber. Each well was filled with 20 μL of sample, and the amperage per gel was set to 15 mA. After electrophoresis, the proteins were transferred to the PVDF membrane. All transfer materials were used from the Transblot Turbo Transfer kit (BioRad, Hercules, CA, USA). After the transfer was completed, the membranes were placed in blocking buffer (PBS containing 2% BSA) for 1 h. 

Next, the blocking buffer was washed three times with PBS + 0.1% Tween20 (PBS-T) solution, for 5 min each time, and then left overnight with the primary antibodies to the markers: ASAHL (1:1000, Santa Cruz, sc-100470), IL-1β (1:1500, ab9722, Abcam, Cambridge, MA, USA), IL-6 (1:1500, ab208113, Abcam, Cambridge, MA, USA), TNFα (1:1000, Thermo Fisher 701135, Thermo Fisher Scientific, Waltham, MA, USA), and α-Tubulin (1:500, ab7291, Abcam, Cambridge, MA, USA) dissolved in PBS buffer with 0.02% BSA (Sigma Aldrich A2153, Missouri, USA) and 0.1% Tween20. After incubation with the primary antibodies, the PBS-T was washed again 3 times for 5 min each, incubated for 1 h with secondary rabbit (Vector Laboratories, PI-1000, Burlingame, CA, USA) and mouse antibodies (Vector Laboratories, PI-2000, Burlingame, CA, USA) dissolved in PBS-T, and then washed again 3 times with phosphate buffer. Chemiluminescence Western Blot ECL Substrate (BioRad, Hercules, CA, USA) was used for detection with an amount of 1 mL of substrate per membrane, and the incubation time was 5 min. Detection was performed using the ChemiDoc gel documentation system (BioRad, Hercules, CA, USA). The obtained images were analyzed using the ImageJ 1.41 software package (NIH, USA).

### 2.9. Statistical Analysis

Further statistical analysis and plotting was performed using GraphPad Prism 8.00 software (GraphPad Software, San Diego, CA, USA). The data are presented as the mean ± SEM (standard error of the mean), and *p* < 0.05, *p* < 0.01, *p* < 0.001 are considered statistically significant. In vivo data were analyzed using two-way ANOVA followed by Tukey’s post hoc multiple comparison analysis. The data obtained from cell cultures were subjected to statistical analysis using one-way ANOVA tests, followed by a post hoc Tukey’s multiple comparison test, and also using Student’s *t*-test.

## 3. Results

### 3.1. OEA-DS Reduced Weight Gain and Decreased Daily Caloric Intake

The results of the body weight measurements show a reduction in weight gain in the groups receiving OEA-DS (in the “CTL” group, the gain was 2.2 ± 0.3 g; in the “CTL + OEA” group, 1.8 ± 0.3 g; in the “DIO” group, 6.8 ± 0.2 g; and in the “DIO + OEA” group, 4.4 ± 0.5 g). Two-way analysis of variance of weight gain revealed a significant effect of obesity (F (1, 20) = 90.06; *p* < 0.0001) and OEA-DS administration (F (1, 20) = 17.07; *p* = 0.0005) and a significant interaction effect of these factors (F (1, 20) = 6.699; *p* = 0.0176), indicating the effect of OEA-DS also on weight gain not only in obese animals but also in the control group receiving the diet supplement (Figure 2a).

Two-way ANOVA of daily caloric intake revealed a significant effect of obesity (F (1, 134) = 228.7; *p* < 0.0001) and OEA-DS administration (F (1, 134) = 7.340; *p* = 0.0076). This result can be interpreted that OEA-DS reduced intake in obesity but did not significantly affect caloric intake in the control animals receiving the diet supplement.

In the animals of the “DIO” group, caloric intake was doubled compared to the controls (10.5 ± 0.3 kcal in the “CTL” group and 21.6 ± 0.9 kcal in the “DIO” group). OEA administration in obese animals reduced the amount of calories consumed by 14% (18.9 ± 0.6 kcal in the “DIO + OEA” group). Also, OEA-DS administration did not significantly reduce calorie intake in the “CTL + OEA” group (9.9 ± 0.3 kcal, which was 5% less compared to the control group). The decrease in calorie intake remained constant throughout the experiment (Figure 2b).

### 3.2. OEA-DS Reduces the Intensity of Morphological Changes and Cell Proliferation in the Spleen

In the “CTL” and “CTL + OEA-DS” groups, the main structural elements of the organ are clearly distinguished: white pulp and red pulp. A slight accumulation of hemosiderin is noted in the red pulp (Figure 3a,b). In diet-induced obesity, the zone of hemosiderin inclusion expanded significantly, and yellow and brown grains were noted in the white pulp, indicating a more intensive process of erythrocyte damage (Figure 3c). Also, in the red pulp of animals from the “DIO” group, a weak diffuse accumulation of leukocytes was noted, and in some places, swelling of the trabeculae was observed. These phenomena were accompanied by the growth of white pulp follicles, which was expressed as a significant increase in the W/R index (1.3 ± 0.1). In the “DIO + OEA-DS” group, lymphoid infiltration was less pronounced, and the accumulation of blood pigment was lower (Figure 3d). The W/R with OEA-DS almost returned to the control level (0.9 ± 0.03) (Figure 3i). Two-way analysis of variance of the white pulp/red pulp ratio revealed significant effects of obesity (F (1, 24) = 58.19; *p* < 0.0001), OEA-DS administration (F (1, 24) = 19.73; *p* = 0.0002), and their interaction (F (1, 24) = 17.59; *p* = 0.0003). Thus, the inflammation developing in obesity promotes intense white pulp overgrowth, and the application of OEA-DS prevents this effect.

The quantification of Ki67-positive cell elements showed a significant increase in the number of proliferating cells in lymphoid follicles in obesity (5389 ± 281.8 cells/mm^3^)—73% higher than in the control group (3115 ± 446.4 cells/mm^3^) and 135% higher than in the “CTL + OEA-DS” group (2292 ± 494.5 cells/mm^3^). The administration of OEA-DS to obese animals promoted a significant decrease in the number of Ki67+ -cells in the white pulp, 68% lower than the control level (1845 ± 440.1 cells/mm^3^) (Figure 3j).

A relevant result was obtained by two-way ANOVA of the number of Ki67-positive cells in lymphoid follicles: significant effects of obesity (F (1, 84) = 8.171; *p* = 0.0054), OEA-DS administration (F (1, 84) = 34.74; *p* < 0.0001), and their interaction (F (1, 84) = 16.92; *p* < 0.0001) were found. 

### 3.3. OEA-DS Administration Was Accompanied by an Increase in PPAR-α Receptor Expression in the Spleen

Two-way analysis of variance for PPAR-a expression revealed a significant effect only for OEA-DS administration in both the white (F (1, 44) = 21.48; *p* < 0.0001) and red spleen pulp (F (1, 40) = 6.182; *p* = 0.0172). Thus, in this case, PPAR-a expression depended only on OEA-DS administration and was independent of obesity (Figure 4i,j).

### 3.4. The Action of OEA-DS Was Accompanied by a Decrease in Inflammatory Reactions at Both the Cellular and Molecular Levels

In the “DIO” group, the development of alimentary-induced obesity was characterized by the formation of the foci of Iba-1-positive macrophage accumulation over the entire area of lymphoid follicles. Collectively, the areas occupied by macrophage accumulation in the tissue occupied an average of 7.9 ± 0.2% of the observed area, which was almost five-fold greater than in the control group (1.7 ± 0.1%) and in the control group receiving OEA-DS (1.6 ± 0.1%). The administration of OEA-DS to obese animals reduced cellular infiltration of the white pulp to the control levels (1.3 ± 0.1%). (Figure 5i). In the red pulp, the density of Iba-1-positive cells in the “DIO” group (9.5 ± 0.4%) was seven times higher than in the “CTL” group (1.5 ± 0.1%) and “CTL + OEA-DS” group (1.7 ± 0.1%), and the values in the “DIO + OEA-DS” group (1.6 ± 0.2%) were almost equal to the control (Figure 6m). In the red pulp, the density of Iba-1-positive cells in the “DIO” group (9.5 ± 0.4%) was seven times higher than in the “CTL” (1.5 ± 0.1%) and “CTL + OEA-DS” (1.7 ± 0.1%) groups; the values in the “DIO + OEA-DS” group (1.6 ± 0.2%) were almost equal to the control (Figure 6m).

Two-way analysis of variance analysis of the area of distribution of Iba1-positive cells revealed a significant effect of obesity (F (1, 88) = 285.1; *p* < 0.0001) and OEA-DS administration (F (1, 88) = 363.8; *p* < 0.0001), and a significant interaction effect of these factors (F (1, 88) = 326.7; *p* < 0.0001) in both white pulp and red pulp: A significant effect of obesity (F (1, 88) =364.6; *p* < 0.0001), OEA-DS application (F (1, 88) = 259.9; *p* < 0.0001), factor interaction (F (1, 88) = 297.4; *p* < 0.0001) was found. These results indicate that OEA-DS prevented the development of obesity-mediated increases in Iba-1 immunoreactivity in all regions of the spleen.

Quantification of the proinflammatory marker CD68 in the “DIO” group in the white (1.4 ± 0.1%) and red (5.9 ± 0.3%) pulp showed its significant increase—more than two-fold compared to the “CTL” groups (0.4 ± 0.4% in the white pulp; 2.4 ± 0.2% in the red pulp) and “CTL + OEA-DS” (0.3 ± 0.03% in the white pulp; 1.8 ± 0.2% in the red pulp). The activation of CD68-positive cellular elements was also observed in the white and red pulp of the “DIO + OEA-DS” group, but with less intensity (0.6 ± 0.5% in the white pulp; 3.5 ± 0.2% in the red pulp) (Figure 5j and Figure 6n).

The “DIO” group showed an increase in CD163-positive areas in the red pulp by 40% (2.8 ± 0.1%) compared to the “CTL” group (2.0 ± 0.1%). The administration of OEA-DS to obese animals increased the stained area by almost two-fold compared to the control (3.6 ± 0.1%) (Figure 6o).

Similar iba-1 results of two-way ANOVA were also found when analyzing the expressions of markers CD68 and CD163. For CD68, significant effects of obesity (F (1, 76) = 139.5; *p* < 0.0001) and OEA-DS application (F (1, 76) = 43.47; *p* < 0.0001), and a significant interaction effect of these factors (F (1, 76) = 38.46; *p* < 0.0001) were found in both the white and red pulp. Significant effects of obesity (F (1, 76) = 117.8; *p* < 0.0001), OEA-DS administration (F (1, 76) = 39.37; *p* < 0.0001), and the interaction of these factors (F (1, 76) = 15.61; *p* = 0.0002) were found.

For CD163, there was a significant effect of obesity (F (1, 80) = 73.79; *p* < 0.0001) and OEA-DS administration (F (1, 80) = 11.87; *p* = 0.0009), and a significant interaction effect of these factors (F (1, 80) = 5.548; *p* = 0.0209). Taken together, the results indicate that OEA-DS administration contributed to the reduction of the intensity of the inflammatory cellular response that developed against the background of obesity.

In the plasma, for TNFα, two-way ANOVA revealed no significant effects of either factor. For IL-1β and IL-6, two-way ANOVA revealed significant effects of obesity (F (1, 8) = 7.462; *p* = 0.0258 for IL-1β; F (1, 8) = 6.943; *p* = 0.0299 for IL-6). A significant effect of OEA-DS administration was found only for IL-6 ((F (1, 8) = 12.08; *p* = 0.0084), as was their interaction effect F (1, 8) = 14.74; *p* = 0.0050). Thus, alimentary-induced obesity enhanced the release of pro-inflammatory cytokines, namely IL-1β and IL-6, into the blood. The administration of OEA-DS to animals with a standard diet did not affect the plasma concentrations of cytokines, and in the case of IL-6, resulted in a significant decrease in obese animals. (Figure 7c).

The results of Western blotting of the spleen tissues also indicate increased expressions of all studied proinflammatory cytokines in the animals of the “DIO” group, several times higher than those in the control group. In the spleen, in the case of TNFα, there was only a significant effect of obesity (F (1, 8) = 19.43; *p* = 0.0023). In contrast, for IL-6 and IL-1β, there were significant effects of both obesity (F (1, 8) = 51.03, *p* < 0.0001 for IL-6; F (1, 8) = 35.11, *p* = 0.0004 for IL-1β) and OEA-DS administration (F (1, 8) = 11.75, *p* = 0.0090 for IL-6; F (1, 8) = 5.567, *p* = 0.0460 for IL-1β ), and their interactions (F (1, 8) = 12.883, *p* = 0.0071 for IL-6; F (1, 8) = 13.03, *p* = 0.0069 for IL-1β). Consequently, obesity affected the levels of all cytokines in the systemic bloodstream, whereby the administration of OEA-DS promoted a decrease in the IL-6 and IL-1β levels, but not TNFα (Figure 7d).

### 3.5. OEA-DS Induced Metabolic Changes in Macrophage Cell Culture

On the macrophage cell line RAW264.7, the MTS test revealed no cytotoxic effect of OEA-DS at concentrations of 0.01, 0.1, 1, and 10 μg/mL (Figure 8a). Macrophage activation by LPS (1 μg/mL) was accompanied by a significant increase in ROS and NO. The addition of OEA-DS to LPS-activated macrophages at concentrations (0.001–10 µg/mL) led to a significant decrease in both ROS (*p* < 0.05) (Figure 8c) and nitric oxide production (Figure 8e). At the same time, the treatment of intact cells with OEA-DS also did not lead to changes in the levels of either ROS (Figure 8d) or nitric oxide (Figure 8b).

The Western blotting data demonstrate that LPS activation made the cytokine production (TNFα, IL-1β, and IL-6) considerably higher. The addition of OEA-DS at concentrations of 1 and 10 μg/mL reduced the production of proinflammatory markers to almost the control levels (Figure 9d). Western blotting also revealed that that macrophages preferentially used the ASAHL enzyme to metabolize OEA-DS. The OEA-DS treatment activated cells to increase the ASAHL synthesis in a dose-dependent manner (Figure 9c).

## 4. Discussion

A significant reduction in caloric intake during OEA-DS administration in obese animals may indicate the involvement of central mechanisms in the regulation of hunger and satiety. This hypothesis is also confirmed by studies [40,41], which showed, by using physiological and biochemical tests on animals, that a decline in caloric intake was not associated with high stress in animals, impaired motor activity, cognitive processes, malaise, pain, or body temperature changes. Also, the study by Romano et al. [42] found no effect of OEA on the modification of eating behavior in animals that did not eat any tasty food during the experimental period. The findings suggest that the effects of OEA-DS are probably directed specifically against overeating by inducing faster satiety, but the exact mechanisms require further study.

The lower body weight gain in both the obese and non-obese animals receiving OEA-DS suggests that the effects of OEA-DS extend not only to excess fat mass but also to the consumption of tasty hypercaloric food. And the presence of significant differences in the morphological profile of the spleen confirms the presence peripheral effects of the dietary supplement, manifested in the reduction of general inflammation and the normalization of metabolic processes by modulating the activity of the immune system cells.

The ability of macrophages to exhibit tissue-specific diversity and to change their phenotype to produce pro- and anti-inflammatory cytokines is well known [43]. The division of macrophage into pro-inflammatory (M1), and alternatively, activated anti-inflammatory (M2) phenotypes was introduced. In turn, one or another pathway of macrophage activation is determined mainly by the local tissue microenvironment, which ensures their constant adaptation to environmental conditions [44,45]. Macrophages, due to their primary involvement in the uptake of pathogen particles, apoptotic cells, or cell debris, have a set of scavenger receptors (SRS), which are able to recognize a wide range of ligands [46].

The transmembrane receptor CD163 belongs to the scavenger receptors and is highly expressed in macrophages [47]. As a rule, CD163 is expressed on the cell surface of alternatively activated M2 macrophages [48]. The presence of anti-inflammatory cytokines, such as IL-4 and IL-10, in the medium is known to induce CD163 expression, whereas the proinflammatory cytokines IL-6, TNFα, IFNγ, and LPS suppress its expression [49]. In the spleen tissue of obese animals treated with OEA-DS, the expression of CD163 is enhanced, which was proven by both the immunohistochemical and Western blotting methods. The in vitro data also show a reduction in the synthesis of pro-inflammatory factors, which occurred simultaneously with an increase in CD163 expression, also confirming the direct effect of OEA-DS on the activation of cells with an anti-inflammatory phenotype. The heightened activity of CD163-positive cells in the DIO group compared to the “CTL” and “CTL + OEA-DS” groups can be explained by the fact that the reparation processes will also take place during the development of inflammation in any case.

CD68 is also considered a member of the scavenger receptor type D (SCARD) family. CD68 is predominantly expressed in the late endosomes and lysosomes of macrophages [50], can be significantly increased in macrophages responding to inflammatory factors [51,52], and is capable of binding apoptotic cells [46,53]. The highest number of CD68-positive elements in the “DIO” group indicates the presence of inflammation and active apoptosis processes in both the white and red pulps, which agrees with the histological data. OEA-DS regulated the ratio of pro- and anti-inflammatory factors in the tissue environment, which, according to the Western blotting of spleen tissue and cell culture, led to a reduction the number of CD68-immunopositive cells. 

Ionized calcium-binding adaptor molecule 1 is selectively expressed by microglia/macrophages [54]. Normally, Iba1 expression is elevated in activated macrophages during inflammation and plays a key part in phagocytosis processes [55]. The abundant activation of macrophages observed in the spleen of animals with alimentary-induced obesity indicates, firstly, the presence of an inflammatory process in the organ and, secondly, the intensive processes of the phagocytosis of destroyed apoptotic cells in both the white and red pulp, which is supported by both the immunoperoxidase reaction results for the other inflammatory markers and the histological staining results.

The proliferation marker Ki67 is expressed in all active phases of the cell cycle (G1, S, and G2), but is absent in resting cells (G0 phase), which makes it an indicator of cell proliferation [56]. The significant increase in the number of Ki67-positive cell elements directly in the lymphoid follicles of the spleen against the background of high levels of proinflammatory cytokines both in the spleen and in the plasma indicates the initiation of proliferation processes of T lymphocytes, induced by inflammation. Proliferating T cells, in turn, also contributed to a secondary increase in proinflammatory cytokine production, which ultimately caused the characteristic cellular pattern of inflammation. The in vivo results are consistent with the in vitro data obtained on the macrophage cell line RAW264.7. The treatment of macrophages with OEA-DS neutralized the LPS-induced activation of the synthesis of the pro-inflammatory factors TNF, IL-1β, and IL-6, and also inhibited the production of ROS and nitric oxide. Other studies also confirm the anti-inflammatory activity of both OEA-DS itself and the other main components of this dietary supplement (PEA, LEA), expressed, in particular, in the reduction of LPS-induced inflammation [25,57,58]. 

N-acylethanolamide acid amidase (NAAA/ASAHL) is the enzyme responsible for the inactivation of NAE to ethanolamine and the corresponding fatty acid [59,60,61]. ASAHL is most highly expressed in macrophages and other cells of the immune system [62]. The dose-dependent increase in the expression of this enzyme in macrophages upon treatment with OEA-DS confirms the cellular uptake of the dietary supplement in the form used.

The elevated numbers of PPAR-positive cells in the spleens of animals treated with OEA-DS without visible differences between the obese and non-obese groups indicates that the increased expression of this receptor was associated specifically with the intake of the substance, and in this case, was not mediated by pathology. A lot of studies confirm the anti-inflammatory effect of PPAR-α agonists, which, in most cases, is mediated by a dysregulation of the transcription factor NFkB [63,64], which leads to the inhibition of the production of proinflammatory factors, such as TNFα, IFN-γ, IL-6, and IL-1β [20,65]. The observed effect of OEA-DS on PPAR-α expression probably underlies the suppression of inflammatory responses at both the molecular and cellular levels.

It is also important to mention the ability of OEA-DS to regulate the local and systemic inflammation that develops in obesity by altering the profile of the gut microbiota [66]. It is known that the microbiota can modulate endocannabinoid activity in the gut [67,68]; moreover, the expression of endogenous OEA-DS in the gut may also depend on the composition of the microbiota [69]. In this way, we can speak about an additional gut microbiota-mediated pathway of inflammation regulation in obesity, which, in turn, requires further study.

Thus, OEA-DS therapy suppresses the synthesis of some of the key proinflammatory cytokines and modulates the activity of immune cells. This is observed in spleen tissue, where the activity of cells with a proinflammatory phenotype is reduced, and, conversely, the mechanisms of cell activation towards an anti-inflammatory phenotype are triggered. The spleen, in this case, is a prime example of how a hypercaloric diet high in cholesterol can not only promote weight gain but can also lead to systemic inflammation. On this basis, the suppression of systemic inflammation in obesity is an important part of comprehensive therapy. And dietary supplements such as OEA-DS, which, in addition to inhibiting weight gain, can also have an anti-inflammatory effect, are promising candidates for the realization of this medical strategy. 

## Figures and Tables

**Figure 1 nutrients-15-04345-f001:**
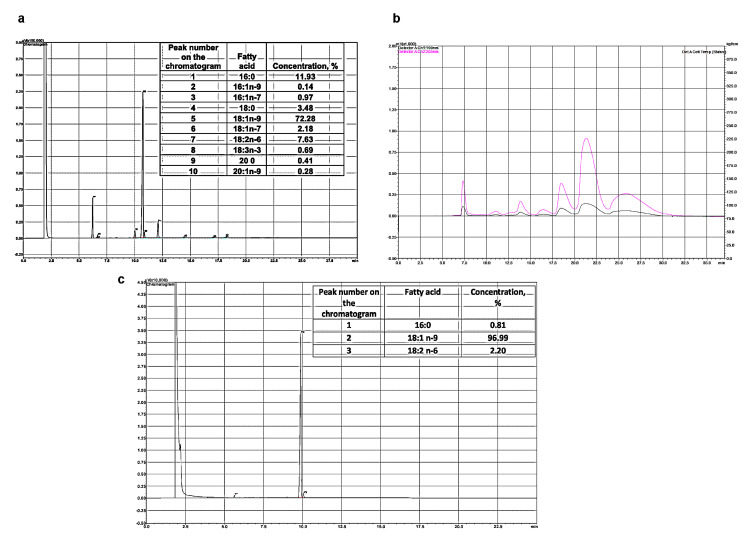
Chromatographic analysis of fatty acid of olive oil (**a**). Peak 1 is palmitic acid, peak 5 is oleic acid, peak 7 is linoleic acid. Chromatogram of preparative extraction of ME oleic acid (HPLC) (**b**). Chromatogram of fatty acid methyl esters analysis (GC) (**c**).

**Figure 2 nutrients-15-04345-f002:**
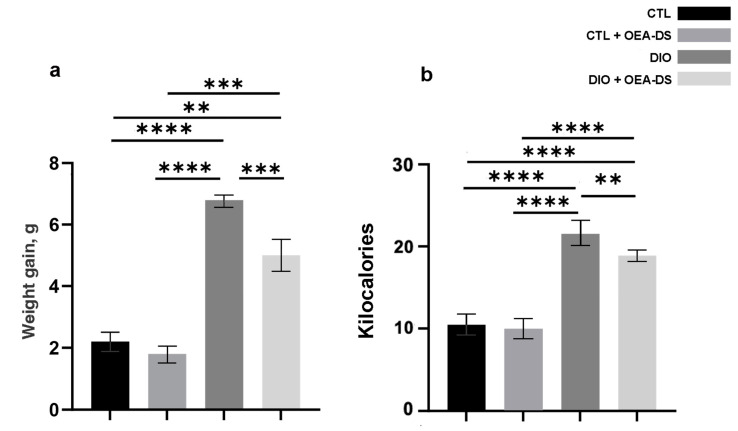
The values of weight gain of experimental animals in the model of alimentary-induced obesity (**a**) and average daily caloric intake per animal (**b**). Data are presented as mean ± SEM, *n* = 12/group, ** *p* < 0.01, *** *p* < 0.001, **** *p* < 0.0001 (two-way ANOVA, post-test Tukey).

**Figure 3 nutrients-15-04345-f003:**
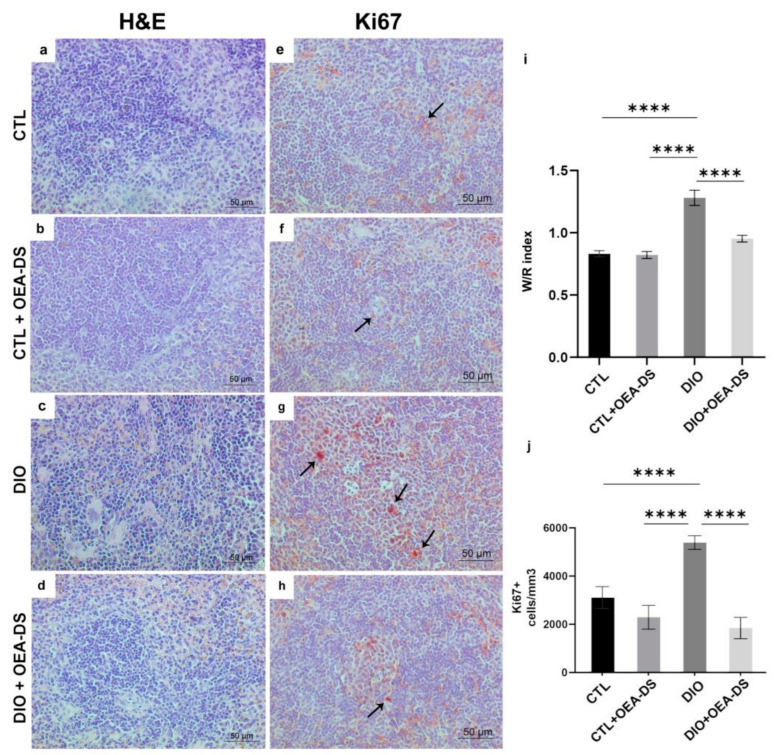
Sections of mouse spleen stained with H&E (**a**–**d**), immunoperoxidase reaction in the white pulp for Ki67 marker (indicated by arrows) (**e**–**h**). Ratio of white to red pulp area in the spleen (**i**). For the Ki67 marker, the number of positively stained cells in one cubic millimeter is compared (**j**). Data are the mean ± SEM, *n* = 6/group, **** *p* < 0.0001 (two-way ANOVA, post hoc Tukey test).

**Figure 4 nutrients-15-04345-f004:**
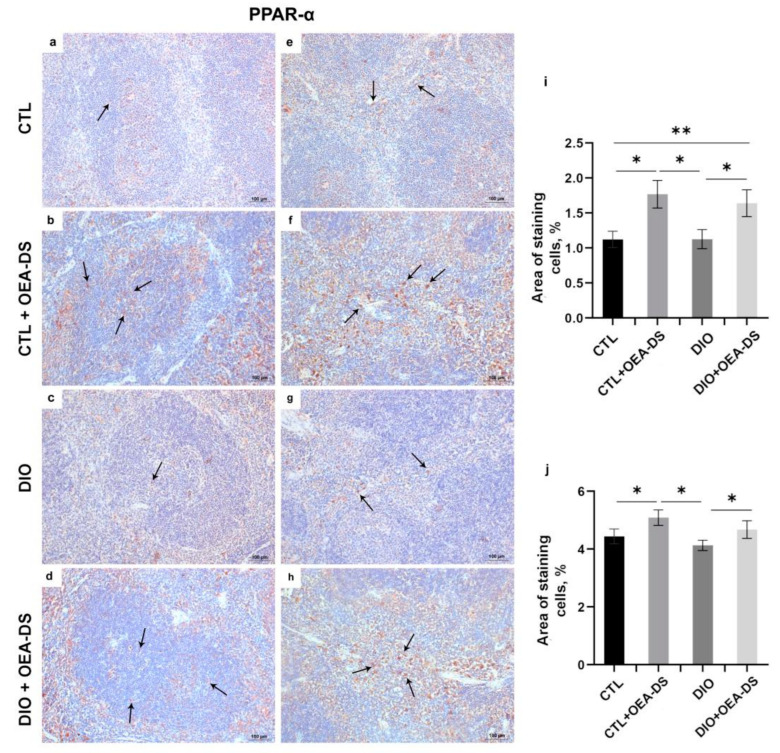
Immunoperoxidase staining for PPAR-α (indicated by arrows) in the white pulp (**a**–**d**) and the red pulp (**e**–**h**). The graphs for white (**i**) and red pulp (**j**) estimate the percentage of immunopositive stained area. Data are the mean ± SEM, *n* = 6/group, * *p* < 0.05, ** *p* < 0.01 (two-way ANOVA, post hoc Tukey test).

**Figure 5 nutrients-15-04345-f005:**
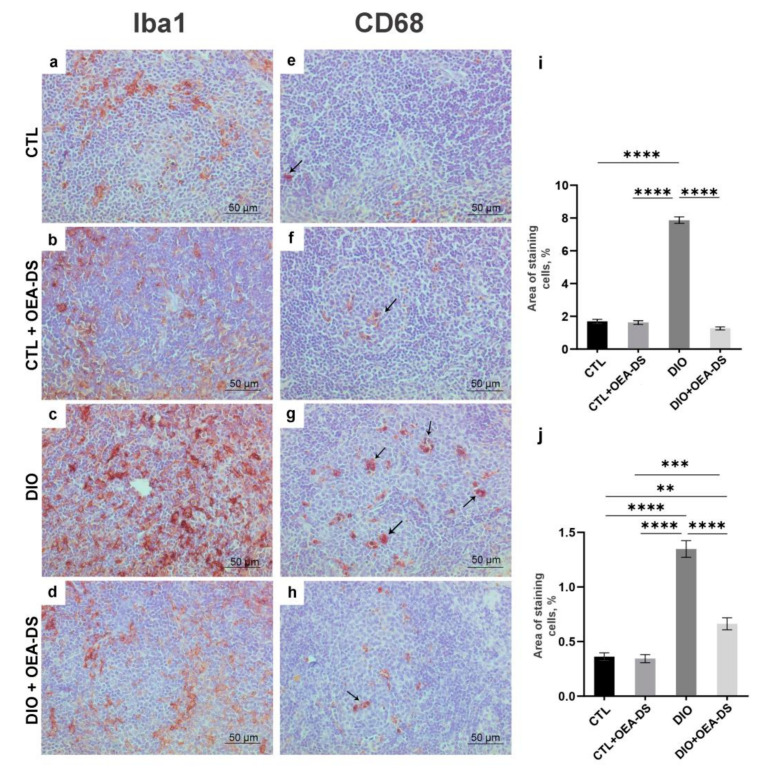
Immunoperoxidase staining in the white pulp for Iba-1 (**a**–**d**) and CD68 (indicated by arrows) (**e**–**h**) markers. The graphs for Iba-1 (**i**) and CD68 (**j**) estimate the percentage of immunopositive stained area. Data are the mean ± SEM, *n* = 6/group, ** *p* < 0.01, *** *p* < 0.001, **** *p* < 0.0001 (two-way ANOVA, post hoc Tukey test).

**Figure 6 nutrients-15-04345-f006:**
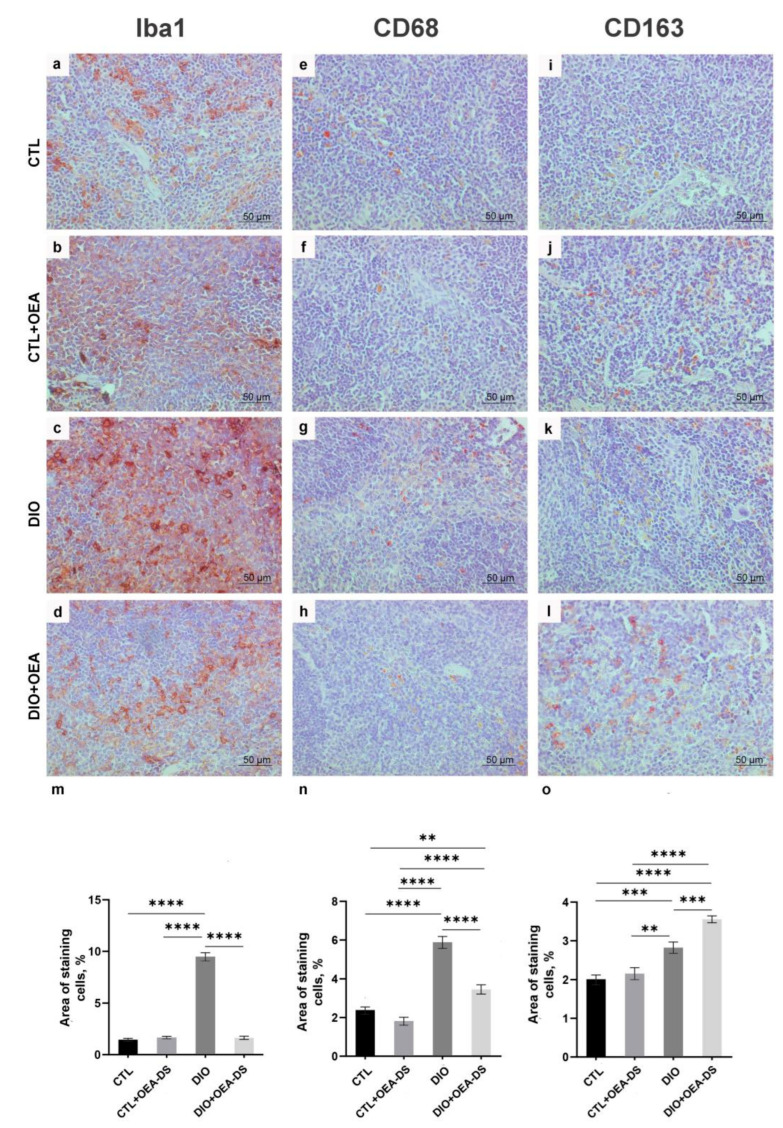
Immunoperoxidase staining in the red pulp for markers Iba-1 (**a**–**d**), CD68 (**e**–**h**), CD163 (**i**–**l**). Percentage of immunopositive stained area is estimated in the graphs (**m**–**o**) for all markers. Data are the mean ± SEM, *n* = 6/group, ** *p* < 0.01, *** *p* < 0.001, **** *p* < 0.0001 (two-way ANOVA, post hoc Tukey test).

**Figure 7 nutrients-15-04345-f007:**
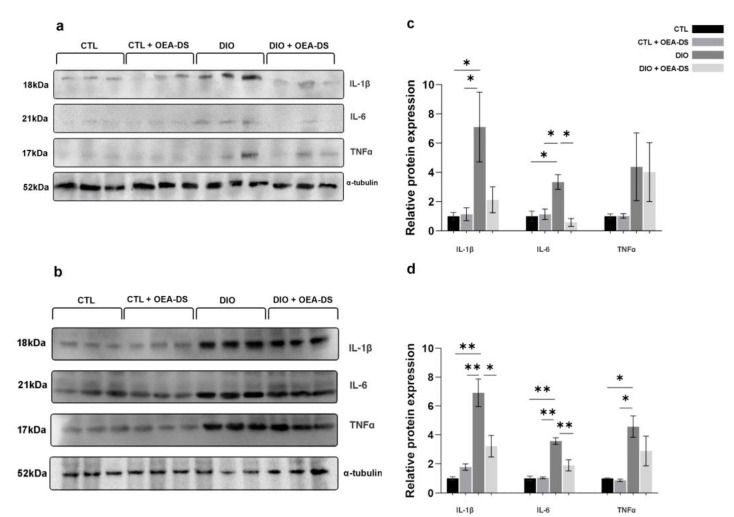
Effects of OEA-DS (200 mg/kg) on cytokine levels in plasma (**a**,**c**) and spleen (**b**,**d**). Expression levels are normalized to α-tubulin and expressed relative to controls. Data are presented as mean ± SEM, * *p* < 0.05, ** *p* < 0.01 (two-way ANOVA, post hoc Tukey test).

**Figure 8 nutrients-15-04345-f008:**
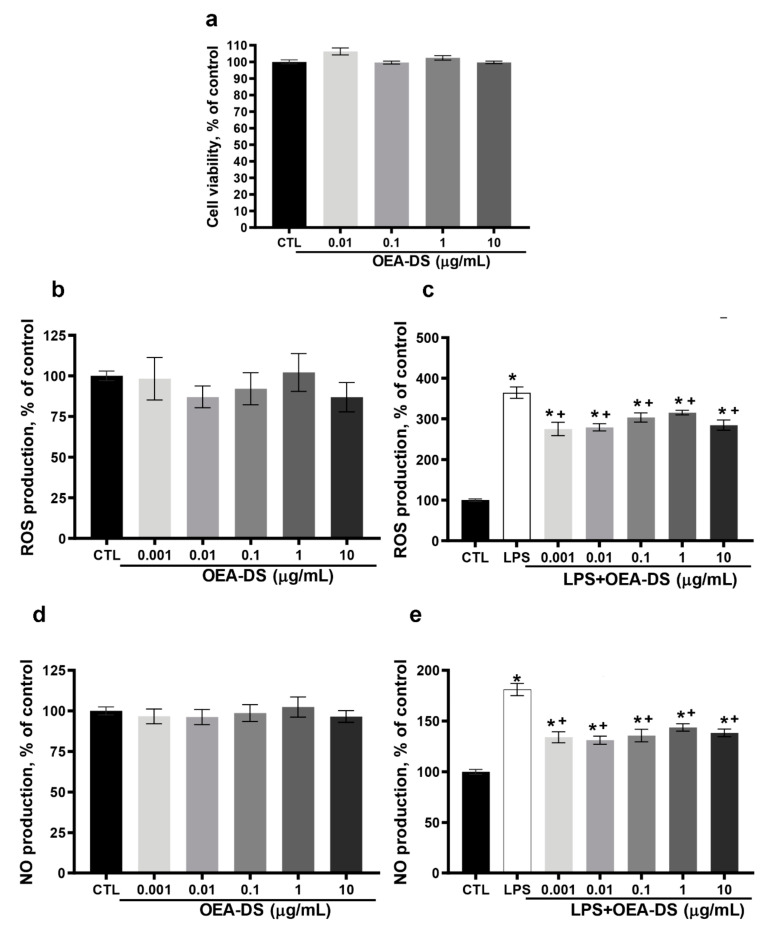
Evaluation of OEA-DS activity in the macrophage cell line RAW264.7. Assessment of cytotoxicity by the MTS test when OEA-DS was added at increasing concentrations (0.1, 1, 10 μg/mL) (**a**). Effects of OEA-DS concentrations on ROS (**b**,**c**) and NO production (**d**,**e**). Data are mean ± SEM, *n* = 9 (number of samples analyzed), * *p* < 0.05—compared with “CTL” ^+^ *p* < 0.05,—compared with “LPS” (Student’s test).

**Figure 9 nutrients-15-04345-f009:**
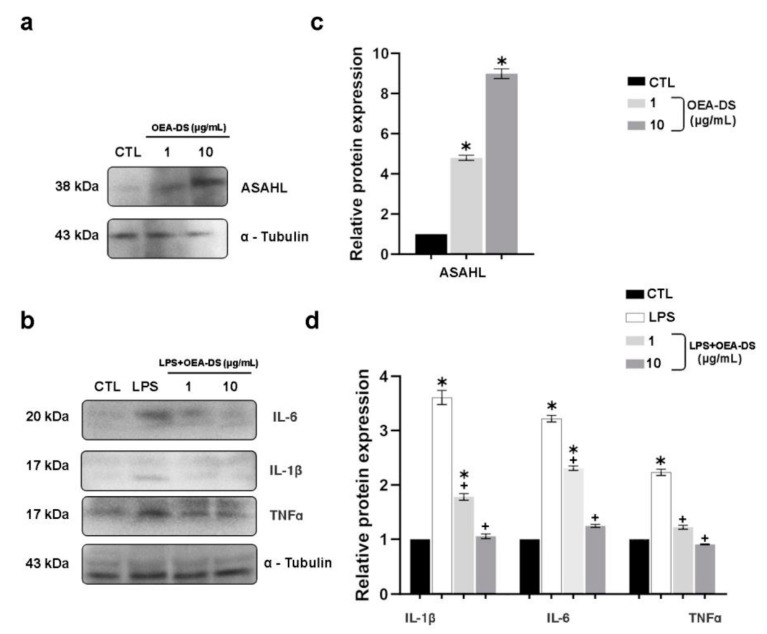
Western blot analysis of the levels, IL-1β, IL-6, and TNFα in RAW264.7 macrophage lysates (**a**,**c**), expression of ASAHL protein in cell culture after the addition of OEA-DS at concentrations of 1 and 10 μg/mL (**b**,**d**). Expression levels are normalized to α-tubulin and expressed relative to controls. For the ASAHL, data are mean ± SEM, *n* = 6/group, * *p* < 0.05 (one-way ANOVA, post hoc Tukey test). For the cytokines, data are mean ± SEM, *n* = 4 (number of repetitions of the experiment), * *p* < 0.05 versus “CTL” ^+^ *p* < 0.05 versus “LPS” (Student’s test).

**Table 1 nutrients-15-04345-t001:** Ratio of macronutrients in the standard feed and the modified feed.

	Standard Feed	Modified Feed
Proteins, g	22.5	13.5
Fats, g	4	22.4
Carbs, g	31.5	38.6
kcal	252	410

## Data Availability

The datasets generated during the current study are available from the corresponding author upon reasonable request.

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
