# Peer review of "Effect of Oleoylethanolamide-Based Dietary Supplement on Systemic Inflammation in the Development of Alimentary-Induced Obesity in Mice"

_nutrients, 2023, doi:10.3390/nu15204345_

Round 1

Reviewer 1 Report

The authors demonstrated the anti-obesity effects and reduction of systemic inflammation in a dietary-induced obesity model in mice using a nutritional supplement based on oleoylethanolamide (OEA-DS).

OEA plays a pivotal role in mediating lipid metabolism, energy homeostasis, and satiety through its action on PPAR-alpha. Understanding the biochemical pathways and mechanisms of OEA can offer insights into the management of metabolic diseases and offer potential avenues for therapeutic interventions.

OEA-DS has the effect of reducing dietary intake and suppressing weight gain in obese animals, and it may also affect the immune system and inflammatory responses.

According to the latest review articles, the anti-obesity effects of oleoylethanolamide and the reduction of inflammatory cytokines are discussed in relation to the function of the gut microbiota. However, in this paper, measurements of the gut microbiota were not conducted, nor was it discussed in the considerations.

The measurement results revealed in this paper can be easily imagined when reading the review articles.

The quality of the experimental data is high, but I was unable to identify the strong points of the research.It might be a good idea to discuss the relationship with the gut microbiota.

What are the strong points of this paper?

Carlotta De Filippo et al. Gut microbiota and oleoylethanolamide in the regulation of intestinal homeostasis. REVIEW article. Front. Endocrinol., 05 April 2023.

Author Response

We thank the reviewer for their interest and time spent on our manuscript.

The main aims of our study were to investigate the complex effect of chronic inflammation developed against the background of hypercaloric diet and obesity on the main target organs. Our study will further analyze other organs as well, including the liver, visceral fat, and hypothalamus. Specifically, this work focuses on changes affecting immune tissue and the assessment of systemic inflammation. Overall, the comprehensive evaluation of organs and changes in their metabolic profile during both the development of obesity and the use of OEA-DS may serve as a source of additional clinical data. We agree that the gut microbiota is an important factor in the development of inflammation, and its assessment could make an important contribution to understanding the mechanisms of obesity-induced inflammation.

But, since we are currently limited in our research methods, we have added this information to the discussion (lines 864-869). However, we will take this comment into account, and in future experiments we will try to outsource animal biomaterial in order to make appropriate measurements.

Reviewer 2 Report

In the study by Ivashkevich and colleagues, the effect of oleoylethanolamide supplement on obesity-related systemic inflammation was investigated. It is unclear the physiological relevance of OEA-DS dose (200mg/kg) used in this study. It seems very high for a supplement or a pharmaceutical therapy. Some clarification is needed.

In the Introduction, the review of the currently available drugs is not complete. Please include GLP-1 mimetics and GLP-1 receptor agonists, which are highly regarded in weight loss treatment and proven to have cardiovascular protections. In addition, time-restricted eating has also been heavily studied and shown beneficial effects on weight loss. 

Please provide the macronutrient information for the control and high fat diet, in the format of % of carbohydrate, fat, and protein in energy density.

In the method, it says “n=10”, however, in the results, it reports “n=12”.

DIO normally refers to diet induced obesity. Please use this standard terminology to describe this group throughout the manuscript.

The statistical method is incorrect. For animal study, two-way ANOVA is needed. Please use the correct method to re-analyse all data sets, which subsequently will change the results and interpretation.

How can the starting body weight of all the mice in the control group be the same? Did the authors deliberately choose the mice with the same body weight?

Why did the authors not use F4/80 to stain for macrophages?

PPAR-α was measured, however, there was no measurement on any of its downstream signals suggesting its anti-inflammatory effects. The authors at least need to provide some discussions on this.

There is no conclusion in this manuscript. Please also provide limitations and future perspectives on this study. 

Line 72: please delete “data” in front of “pathway”.

Line 74: what are “acute phase proteins”?

Some expressions are odd. Proof reading is recommended.

Author Response

We thank the reviewer for his attention and interest in our manuscript.

Below there are the answers to your questions and comments, point-by-point.

In the study by Ivashkevich and colleagues, the effect of oleoylethanolamide supplement on obesity-related systemic inflammation was investigated. It is unclear the physiological relevance of OEA-DS dose (200mg/kg) used in this study. It seems very high for a supplement or a pharmaceutical therapy. Some clarification is needed.

  • Most studies used parenteral administration of OEA, but since we had animals receiving OEA with their feed we focused on studies where it was administered orally: doi.org/10.1016/j.phrs.2003.12.006, doi: 10.1194/jlr.M013391, DOI 10.1194/jlr.C300008-JLR200, https://doi.org/10.1016/j.jnutbio.2009.07.006. In these studies, food intake was significantly reduced generally at the highest doses, i.e., 100 and 200 mg/kg.

In the Introduction, the review of the currently available drugs is not complete. Please include GLP-1 mimetics and GLP-1 receptor agonists, which are highly regarded in weight loss treatment and proven to have cardiovascular protections. In addition, time-restricted eating has also been heavily studied and shown beneficial effects on weight loss.

  • Thanks for the comment, the mentioned information has been added to the introduction (lines 31-37, 42-46).

Please provide the macronutrient information for the control and high fat diet, in the format of % of carbohydrate, fat, and protein in energy density.

  • A table of macronutrients has been added to Materials and Methods chapter.

In the method, it says “n=10”, however, in the results, it reports “n=12”.

  • Thank you for the comment, the typo has been corrected.

DIO normally refers to diet induced obesity. Please use this standard terminology to describe this group throughout the manuscript.

  • Corrected

The statistical method is incorrect. For animal study, two-way ANOVA is needed. Please use the correct method to re-analyse all data sets, which subsequently will change the results and interpretation.

  • Corrected

How can the starting body weight of all the mice in the control group be the same? Did the authors deliberately choose the mice with the same body weight?

  • You are right, the animals were the same weight and it wasn't our deliberate choice. However, due to a change in the statistical analysis method, an error appeared in the graph. Now the graphs show the average gain, not the initial and final weight of the mice.

Why did the authors not use F4/80 to stain for macrophages?

  • All methods used in this work were chosen based on the availability of reagents, no analysis was performed due to our lack of relevant reagents.

PPAR-α was measured, however, there was no measurement on any of its downstream signals suggesting its anti-inflammatory effects. The authors at least need to provide some discussions on this.

  • Your point of view is fair, but the purpose of this work was not to study the biochemistry and molecular biology of the PPARa signaling pathways. Our study was more focused on finding and identifying the final effector molecules of the executors that mediate the cellular response and determine the levels of both systemic inflammation and individual target organs of obesity. In addition, we focused on readily available experimental data indicating that the use of PPARa agonists, by increasing levels of the corresponding receptor, decreases NF-kB levels (doi: 10.1371/journal.pone.0028502, doi: 10.1074/jbc.273.49.32833, doi: 10.1074/jbc.M004045200, doi: 10.1161/01.cir.99.24.3125). We did not perform relevant measurements due to lack of reagents, but we can include relevant information in the discussion if required.

There is no conclusion in this manuscript. Please also provide limitations and future perspectives on this study.

  • Thanks for the comment, the conclusion has been added (lines 870-882).

Line 72: please delete “data” in front of “pathway”.

  • Corrected

Line 74: what are “acute phase proteins”?

  • Corrected